# SEMANTIC CODE REPAIR USING NEURO-SYMBOLIC TRANSFORMATION NETWORKS

## ABSTRACT

We study the problem of *semantic code repair*, which can be broadly defined as automatically fixing non-syntactic bugs in source code. The majority of past work in semantic code repair assumed access to unit tests against which candidate repairs could be validated. In contrast, the goal here is to develop a strong statistical model to accurately predict both bug locations and exact fixes *without* access to information about the intended correct behavior of the program. Achieving such a goal requires a robust contextual repair model, which we train on a large corpus of real-world source code that has been augmented with synthetically injected bugs. Our framework adopts a two-stage approach where first a large set of repair candidates are generated by rule-based processors, and then these candidates are scored by a statistical model using a novel neural network architecture which we refer to as *Share, Specialize, and Compete*. Specifically, the architecture (1) generates a *shared* encoding of the source code using an RNN over the abstract syntax tree, (2) scores each candidate repair using *specialized* network modules, and (3) then normalizes these scores together so they can *compete* against one another in comparable probability space. We evaluate our model on a real-world test set gathered from GitHub containing four common categories of bugs. Our model is able to predict the exact correct repair 41% of the time with a single guess, compared to 13% accuracy for an attentional sequence-to-sequence model.

## 1 INTRODUCTION

The term *automatic code repair* is typically used to describe two overarching tasks: The first involves fixing *syntactic errors*, which are malformations that cause the code to not adhere to some language specification (Gupta et al., 2017; Bhatia and Singh, 2016). The second, which is the focus of this work, involves fixing *semantic bugs*, which refer to any case where the actual program behavior is not the same as the behavior the programmer intended. Clearly, this covers an extremely wide range of code issues, so this work is limited to a class of *semantic bugs*, which we roughly define as: "Bugs that can be identified and fixed by an experienced human programmer, without running the code or having deep contextual knowledge of the program." This does not imply that the bugs are trivially fixable, as they often require time-consuming analysis of the code, rich background knowledge of the language and APIs, and complex logical reasoning about the original programmer's intent.

Unlike previous work, we do not assume access to unit tests at training or test time. This requirement is important because it forces development of models which can infer intended semantic purpose from source code before proposing repairs, as a human programmer might. Most previous work relies on unit tests – a common theme is combining coarse-grained repair models with search algorithms to find some repair that satisfies unit tests (Harman, 2010; Singh et al., 2013). In contrast, our proposed task requires models to deeply understand the code in order to propose a single set of repairs. Thus, semantic code repair without unit tests presents a concrete, real-world test bed for the more general task of understanding and modifying source code.

Our semantic repair model was trained on a large corpus of open-source Python projects with synthetically injected bugs. We test on both real-bug and synthetic-bug test sets. [1] To train the repair model, we first evaluated an attentional sequence-to-sequence architecture. Although this model was able to achieve non-trivial results, we believe it to be an unsuitable solution in a number of ways, such as the lack of direct competition between repair candidates at different locations. Instead, we

---

[1] All data sets will be made publicly available.

use an alternative approach which decouples the non-statistical process of generating and applying repair proposal from the statistical process of scoring and ranking repairs.

This two-stage process itself is not new, but the core novelty in this work is the specific neural framework we propose for scoring repair candidates. We refer to our architecture as a *Share, Specialize, and Compete (SSC)* network:

- **SHARE**: The input code snippet is encoded with a neural network. This is a *shared* representation used by all repair types.
- **SPECIALIZE**: Each repair type is associated with its own *specialized* neural module (Andreas et al., 2016), which emits a score for every repair candidate of that type.
- **COMPETE**: The raw scores from the specialized modules are normalized to *compete* in comparable probability space.

Our model can also be thought of as an evolution of work on neural code completion and summarization Allamanis et al. (2016); Bhoopchand et al. (2016). Like those systems, our SHARE network is used to learn a rich semantic understanding of the code snippet. Our SPECIALIZE modules then build on top of this representation to learn how to identify and fix specific bug types.

Although we have described our framework in relation to the problem of code repair, it has a number of other possible applications in sequence transformation scenarios where the input and output sequences have high overlap. For example, it could be applied to natural language grammar correction (Schmaltz et al., 2016), machine translation post editing (Libovický et al., 2016), source code refactoring (Allamanis et al., 2015), or program optimization (Bunel et al., 2016).

## 2 RELATED WORK

We believe this paper to be the first work addressing the issue of semantic program repair in the absence of unit tests, where functionality must be inferred from the code. However, our work adds to a substantial literature on program repair and program analysis, some of which we describe below:

**Neural Syntax Repair:** There have been several recent techniques developed for training neural networks to correct *syntax* errors in code. DeepFix (Gupta et al., 2017) uses an attentional seq-to-seq model to fix syntax errors in a program by predicting both the buggy line and the statement to replace it. Bhatia and Singh (2016) train an RNN based token sequence model to predict token insertion or replacement at program locations provided by the compiler to fix syntax errors.

**Statistical Program Repair:** Approaches such as Arcuri and Yao (2008) and Goues et al. (2012) use genetic programming techniques to iteratively propose program modifications. Prophet (Long and Rinard, 2016) learns a probabilistic model to rank patches for null pointer exceptions and array out-of-bounds errors. The model is learnt from human patches using a set of hand-engineered program features. In contrast, our neural model automatically learns useful program representations for repairing a much richer class of semantic bugs.

**Natural Source Code / Big Code:** A number of recent papers have trained statistical models on large datasets of real-world code. These papers study tasks involving varying degrees of reasoning about source code, such as code completion (Raychev et al., 2015; 2014; Bhoopchand et al., 2016) and variable/class/function renaming (Raychev, 2016; Allamanis et al., 2015).

**Rule-Based Static Analyzers:** Rule-based analyzers for Python (`Pylint` (Thenault, 2001) and `Pyflakes` (PyCQA, 2012)) handle a highly disjoint set of issues compared to the type of bugs we are targeting, and generally do not directly propose fixes.

## 3 PROBLEM OVERVIEW

As mentioned in the introduction, our goal is to develop a system which can statically analyze a piece of code and predict the location of the bug along with the actual fix. We do not assume to have unit tests or any other specification associated with the snippet being repaired. These proposed repairs can be directly presented to the user, or taken as input to some downstream application. Since the task of "fixing bugs in code" is incredibly broad, we limit ourselves to four classes of common Python bugs that are described with examples in Section 3.

Ideally, we would train such a repair model using a large number of buggy/repaired code snippets. However, such a large data set does not exist. It *is* possible to extract a modest test set of genuine bugs from project commit histories, but it is not enough to train a large-scale neural network. Fortunately, there is a large amount of real-world *non-buggy* code available to which bugs can be injected. We demonstrate that a model trained on synthesized bugs is able to generalize to a test set with real bugs.

**Training Data**  To create the training data, we first downloaded all Python projects from GitHub that were followed by at least 15 users and had permissive licenses (MIT/BSD/Apache), which amounted to 19,000 total repositories. We extracted every function from each Python source file as a *code snippet*. In all experiments presented here, each snippet was analyzed on its own without any surrounding context. All models explored in this paper only use static code representations, so each snippet must be *parsable* as an Abstract Syntax Tree (AST), but does not need to be *runnable*. Note that many of the extracted functions are member functions of some class, so although they can be parsed, they are not runnable without external context. We only kept snippets with between 5 and 300 nodes in their AST, which approximately corresponds to 1 to 40 lines of code. The average extracted snippet had 45 AST nodes and 6 lines of code.

This data was carved into training, test, and validation at the repository level, to eliminate any overlap between training and test. We also filtered out any training snippet which overlapped with any test snippet by more than 5 lines. In total we extracted 2,900,000 training snippets, and held-out 2,000 for test and 2,000 for validation.

**Bug/Repair Types**  In this work, we consider four general classes of semantic repairs, which were chosen to be "simple" but still common during development, as reported by the Python programmers:

- **VarReplace**: An incorrect local variable is used at a particular location, and should be replaced with another variable from the snippet.
- **CompReplace**: An incorrect comparison operator is used at a particular location.
- **IsSwap**: The `is` operator is used instead of `is not`, or vice versa.
- **ClassMember**: A `self` accessor is missing from a variable.

*VarReplace*

```
def send(self, stream, msg, idt):
  to_send = []
  if isinstance(idt, bytes):
    idt = [idt]
  if (idt is not None):
    to_send.extend(idt)
  to_send.append(self.sign(msg))
  to_send.extend(msg)
  stream.send( msg → to_send )
```

*CompReplace*

```
def derive(self, x, is_even):
  c = ecdsa.SECP256k1.curve
  (a, b, p) = (c.a(), c.b(), c.p())
  alpha = ((pow(x, 3, p)) + a*b)
  beta = ecdsa.srmp(alpha, p)
  if ((beta % 2) != → == is_even):
    beta = (p - beta)
  return beta
```

*IsSwap*

```
def action(self, event, plaw):
  if self.subtracted:
    self.unsubtract()
  if ('nwidths' in kwargs):
    kwargs.pop('nwidths')
  if (plaw is not → is None):
    plaw = self.plaw
```

*ClassMember*

```
def __init__(self, form):
  self.transport = 'udp'
  self.ptime = 20
  self.provisioning = False
  mode → self.mode = 'manual'
  for var in form:
    setattr(self, var, form[var])
```

Generating synthetic bugs from these categories is straightforward. For example, for *VarReplace*, we synthesize bugs by replacing one random variable from a snippet with another variable from the same snippet. All bug types, locations, and replacements were chosen with random uniform likelihood. We applied this bug synthesis procedure to all of the training snippets to create our training data, as well as a synthetic test set (*Synth-Bug Test*).

**Real-Bug Test Set**  In order to evaluate on a test set where both the code *and* bugs were real, we mined the Git commit history from the projects crawled from Github. We found that it was quite difficult to automatically distinguish bug repairs from other code changes such as refactoring, especially since we wanted to avoid introducing biases into the data set through the use of complex filtering heuristics. For this reason, we limited extraction to commits where exactly one line in a file

was changed, and the commit contained a word from the list "bug, error, issue, exception, fix". We then filtered these commits to only keep those that correspond to one of our four bug types. Overall, we obtained 926 buggy/repaired snippet pairs with exactly one bug each. We believe that the small number of extracted snippets does not reflect the true frequency of these bugs during development, but rather reflect the fact that (1) one line Git commits are quite rare, (2) these type of bugs rarely make it into the public branch of high-quality repositories.

## 4 BASELINE ATTENTIONAL SEQUENCE-TO-SEQUENCE MODEL

Since the goal of program repair is to transform a buggy snippet into a repaired snippet, an obvious baseline is an attention sequence-to-sequence neural network (Bahdanau et al., 2014), which has been successfully used for the related tasks of syntatic code repair and code completion. On those tasks, sequence-to-sequence models have been shown to outperform a number of baseline methods such as n-gram language models or classifiers.

Because this model must actually generate a sequence, we first converted the buggy/repaired ASTs from Synth-Bug Train back to their tokenized source code, which is a simple deterministic process. The architecture used is almost identical to the machine translation system of Bahdanau et al. (2014). To handle the high number of rare tokens in the data, tokens were split by underscores and camel case. The size of the neural net vocabulary was 50,000, and the final out-of-vocabulary (OOV) rate was 1.1%. In evaluation we included OOVs in the reference, so OOVs did not cause a degradation in results. The LSTMs were 512-dimensional and decoding was performed with a beam of 8. When evaluating on the Single-Repair Synth-Bug Test set, the 1-best output exactly matches the reference 26% of the time. If we give the model credit when it predicts the correct repair but also predicts other changes, the accuracy is 41%.

Although this accuracy seem to be non-trivial, there are some intuitive weaknesses in using a sequence-to-sequence architecture for semantic code repair. First, the system is burdened with constructing the entire output sequence, even though on average it is 98.5% identical to the input. Second, potential repairs at different locations do not fairly "compete" with one another in probability space, but only compete with tokens at the same location. Third, it is difficult to use a richer code representation such as the AST, since the repaired code must be generated.

## 5 SHARE, SPECIALIZE, AND COMPETE (SSC) MODEL

Instead of directly generating the entire output snippet with a neural network, we consider an alternative approach where repairs are iteratively applied to the input snippet. Here, for each bug type described in Section 3, the system proposes all possible repairs of that type in the snippet. Although these candidate generators are manually written, they simply propose all possible repairs of a given type and do not perform any heuristic pruning, so each of the four generators can be written in a few lines of code. The challenging work of determining the correct repair using the code context is performed by our statistical model.

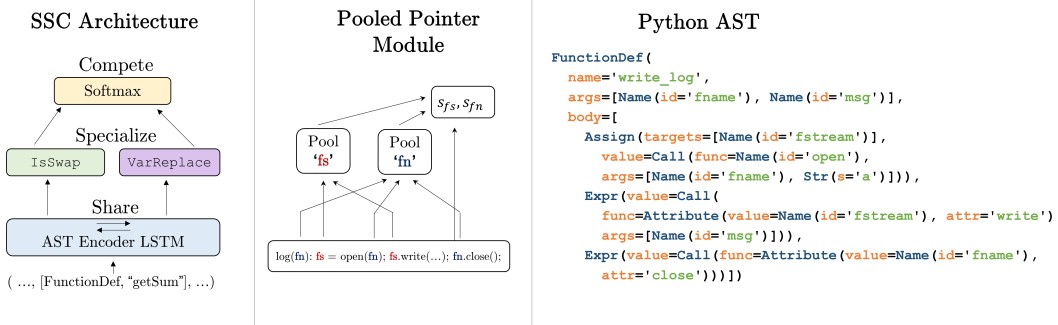

Figure 1: **Model Visualization**: A visualization of the *Share, Specialize, and Compete* architecture for neural program repair.

For clarity of terminology, a *repair candidate* is a particular fix that can be made at a particular location (e.g., "Replace == with != at node 4"). A *repair instance* refers to a particular repair location the generator proposes and *all* of the candidates at that location. Each instance is guaranteed to have exactly one *no-op candidate*, which results in no change to the AST if applied (e.g., "Replace == with == at node 4"). The *reference label* refers to the correct candidate of a given instance (e.g., "The correct replacement at node 4 is <="). Note that for the majority of repair instances that are proposed, the reference label will be the no-op candidate.

We now present the statistical model used to score repair candidates. We refer to it as a *Share, Specialize, and Compete (SSC)* network. A visual representation is given in Figure 1.

## 5.1 SHARE

The SHARE component performs a rich encoding of the input AST using a neural network. Crucially, this encoding is only conditioned on the AST itself and not on any repair candidates, so it serves a *shared* representation for the next component. This network can take many forms, with the only restriction being that it must emit one vector of some dimension $d$ for each node in the AST. An example of a Python AST is given on the right side of Figure 1.

Here, for efficiency purposes, we encode the AST with a sequential bidirectional LSTM by enumerating a depth first traversal of the nodes, which roughly corresponds to "source code order." However, we encode the rich AST structure by using embeddings for (1) the absolute position of the node in the AST, (2) the type of the node in the AST, (3) the relationship between the node and its parent, and (4) the surface form string of the node.

These tokens are projected through an embedding layer and then concatenated, and the resulting vector is used as input to a bidirectional LSTM. The output of this layer is represented as $H = (h_1, h_2, ..., h_n)$, where $h_i \in \mathbb{R}^d$, $d$ is the hidden dimension, and $n$ is the number of nodes in the AST.

The core concept of the *shared* component is that the vast majority of neural computation is performed here, independent of the repairs themselves. We contrast this to an alternative approach where each repair candidate is applied to the input AST and each resulting repair candidate AST is encoded with an RNN – such an approach would be orders of magnitude more expensive to train and evaluate.

## 5.2 SPECIALIZE

The SPECIALIZE component scores each repair candidate using a specialized *network module* (Andreas et al., 2016) for each repair type. Instances of the same type are processed by the same module, but obtain separate scores since they have different input. Each module takes as input the shared representation $H$ and a repair instance $R$ with $m$ candidates. It produces an *un-normalized* scalar score for each candidate in the instance, $\hat{s} = (s_1, ..., s_m)$. We use two module types:

**Multi-Layer Perceptron (MLP) Module**: This module performs scoring over a fixed label set using one non-linear hidden layer. This is used for the *CompReplace*, *IsSwap*, and *ClassMember* generators. It is computed as:

$$\hat{s} = V \times \tanh(Wh_j)$$

where $V \in \mathbb{R}^{m \times c}$, $W \in \mathbb{R}^{c \times d}$, $c$ is the hidden dimension, $m$ is the number of labels (i.e., transform candidates), and $j$ is the transform location corresponding to the transform instance $T$. Note that separate $V$ and $W$ weights are learned for each repair type.

**Pooled Pointer Module**: Predicting variables for *VarReplace* presents a unique challenge when modeling code repair. First, the variable names in a test snippet may never have been seen in training. More importantly, the semantics of a variable are primarily defined by its usage, rather than its name. To address this, instead of using a fixed output layer, each candidate (i.e., another variable) is encoded using *pointers* to each usage of that variable in the AST. An example is given in Figure 1. Formally, it is computed as:

$$s_i \quad = \quad \tanh(Wh_j) \cdot [\mathrm{MaxPool}_{k \in p_i}(\tanh(Vh_k))]$$

where $i$ is the candidate (i.e., variable) index, $p_i$ is the list of locations (pointers) of the variable $i$ in the AST, $j$ is the location of the repair in the AST, and $V, W \in \mathbb{R}^{c \times d}$ are learned weight matrices.

## 5.3 COMPETE

Once a scalar score has been produced for each repair candidate, these must be normalized to *compete* against one another. We consider two approaches to normalizing these scores:

**Local Norm**: A separate softmax is performed for each repair instance (i.e., location and type), so candidates are only normalized against other candidates in the same instance, including no-op. At test time we sort all candidates across all instances by probability, even though they have not been normalized against each other.

**Global Norm**: All candidates at all locations are normalized with a single softmax. No-op candidates are *not* included in this formulation.

## 6 EXPERIMENTAL RESULTS

We train the SSC model on the Synth-Bug Train data for 30 epochs. Different bugs are synthesized at each epoch which significantly mitigates over-fitting. We set the hidden dimensions of the SHARE and SPECIALIZE components to 512, and the embedding size to 128. A dropout of 0.25 is used on the output of the SHARE component. Training was done with plain SGD + gradient clipping using an in-house toolkit. A small amount of hyperparameter tuning was performed on the Synth-Bug Val set.

In the first condition we evaluate, all snippets in both training and test have exactly one bug each. As was described in Section 3, for Synth-Bug Test, the code snippets are real, but the bugs have been artificially inserted at random. For Real-Bug Test, we extracted 926 buggy/fixed snippet pairs mined from GitHub commit logs, so both the snippet and bug are real. The average snippet in the Real-Bug Test set has 31 repair locations and 102 total repair candidates, compared to 20 locations and 50 candidates of the Synth-Bug Test test set.

Table 1 presents Single-Repair results on Synth-Bug and Real-Bug test sets. The accuracy metric denotes how often the 1-best repair prediction exactly matches the reference repair, i.e., the model correctly detects where the bug is and correctly predicts how to fix it. In this case, the model was constrained to predict exactly one repair, but all candidates across all repair types are directly competing against one another. On Synth-Bug, the SSC model drastically outperforms the attentional sequence-to-sequence model, even using the upper bound seq-to-seq accuracy. Since global normalization and local normalization have similar performance and it is not obvious how to extend global normalization to multiple repairs, we use local normalization for multi-repair experiments.

On Real-Bug Test, the absolute accuracy is lower than on Synth-Bug Test, but the SSC model still significantly outperforms the seq-to-seq baseline. To better understand the absolute quality of the Real-Bug Test results, we perform a preliminary human evaluation in Section 6.

| Single-Repair | | | | Multi-Repair | | |
|---|---|---|---|---|---|---|
| | **Synth-Bug Accuracy** | **Real-Bug Accuracy** | | **Num Bugs** | **Synth-Bug** | |
| | | | | | **F-Score** | **Exact Accuracy** |
| Att. Seq-to-Seq | 26% (40%[†]) | 13% (18%[†]) | | 0 | - | 82% |
| SSC (Global Norm) | 86% | 41% | | 1 | 85% | 78% |
| SSC (Local Norm) | 87% | 41% | | 2 | 84% | 61% |
| *VarReplace* | 82% | 36% | | 3 | 81% | 45% |
| *CompReplace* | 80% | 29% | | All | 82% | 66% |
| *IsSwap* | 96% | 82% | | | | |
| *ClassMember* | 95% | 56% | | | | |

Table 1: **Repair Accuracy**: 1-best repair accuracy prediction for the single-repair and multi-repair condition [†]Denotes "upper bound" accuracies as in Sec. 4.

Example predictions from the Real-Bug Test set are presented below. The red region is the bug, and the green is the reference repair. For the incorrect predictions, the blue region is the predicted repair. Results on all 926 Real-Bug Test examples are provided in the supplementary material.

## Correct Predictions

```python
def start(nova_client, start_retry_interval, key_path):
    server = get_server_by_context(nova_client)
    if is_external_resource(ctx):
        ctx.logger.info('Validating external server')
        if (server.status != SERVER_STATUS_ACTIVE):
            raise NonRecoverableError()
        return
    if (server.status == SERVER_STATUS_ACTIVE):
        ctx.logger.info('Server is {0}'.format(server.status))
        if ctx.node.properties['use_password']:
            key = _get_private_key(key_path)
            password = nova_client → server .get_password(key)
```

```python
def val_addr(self, buf, af, *args):
    if (af == AFI_T['IPv4']):
        m = 4
        _af = socket.AF_INET
    elif (af == AFI_T['IPv6']):
        m = 16
        _af = socket.AF_INET6
    else:
        n = -1
    n = m if (len(args) == 0) else ((args[0])
    if ((n <= → < 0) or ((len(buf)) < n)):
        return None
```

```python
def wcswidth(pwcs, n):
    if (n is not → is None):
        end = len(pwcs)
    else:
        end = n
    idx = slice(0, end)
    width = 0
    for char in pwcs[idx]:
        wcw = wcwidth(char)
        width += wcw
    return width
```

## Incorrect Predictions

```python
def _get_python_files(paths):
    for path in paths:
        if os.path.isdir(path):
            for (base, dirs, files) in os.walk(path):
                for file in files:
                    fullpath = os.path.join(base, file)
                    if pep8style.excluded(fullpath):
                        yield fullpath
        elif pep8style.excluded( fullpath → path ):
            yield path → fullpath
```

```python
def purge_kernels(self, msg_content):
    failures = []
    for kernel_id in self.km._kernels:
        success = self.km.kill_kernel(kernel_id)
        if (not success):
            failures.append(kernel_id)
    reply = {'status': 'All kernels killed!'}
    success = (len(failures) > → == 0)
    if (not success):
        reply['status'] = failures → success
    return self._form_message(reply)
```

```python
def check_token(self, user, token):
    (ts_b36, hash) = token.split('-')
    ts = base36_to_int(ts_b36)
    tok = self._make_token(user, ts)
    if (tok != token → ts ):
        return False
    today = self.today()
    if (td - ts > TIMEOUT → self.TIMEOUT ):
        return False
```

In the multi-repair setting, we consider the more realistic scenario where a snippet may have multiple bugs, or may have none. To model this scenario, the data was re-generated so that 0, 1, 2, or 3 bugs was added to each training/test/val snippet, with equal probability of each. We refer to these new sets as *Synth-Multi-Bug Test* and *Synth-Multi-Bug Val*. Unfortunately, we were not able to extract multi-bug examples from the Real-Bug data.

The major new complexity is that the system must now determine *how many* repairs to predict per snippet, if any. We use a simple threshold-based approach: Since each repair candidate is assigned a probability by the model, we simply emit all repairs which have probability greater than $\delta$. The system is *not* constrained to emit only 3 repairs. A parameter sweep over the validation set revealed that accuracy is surprisingly un-sensitive to $\delta$, so we simply use $\delta = 0.5$. Note that we only perform a single pass of repair scoring here, but in future work we will explore an iterative decoder.

Results are presented on the right side of Table 1. For accuracy at the per-repair level, there is only a moderate decrease in F-score from 85% to 81% between the 1-repair and 3-repair settings. The Exact Accuracy does decrease significantly, but not beyond the "expected value." In other words, three independent 1-repair snippets have an expected accuracy of $0.78^3 = 0.47$, which is similar to the 45% accuracy observed for 3-repair snippet. We also see that the system is 82% accurate at correctly predicting when a snippet has *no* bugs.

**Human Evaluation** To better understand the significance of the performance of our system, we performed a preliminary human evaluation under identical conditions to the model. The evaluator was presented with a snippet from the test set, where all repair instances were highlighted in the code. The evaluator could click on a repair location to see all candidates at that location. They were explained each of the four bug types and told that there was always exactly one bug per snippet. This evaluation required experienced Python programmers performing a complex task, so we performed a small evaluation using 4 evaluators and 30 snippets each from the Real-Bug Test set. Evaluators typically used 2-6 minutes per snippet. These snippets were limited to 150 nodes for the benefit of the human evaluators, so the SSC model accuracy is higher on this subset than on the full set.

On these snippets, the humans achieved 37% accuracy compared to the 60% accuracy of the SSC model. One possible reason for this performance gap is that the model is simply better than humans at this task, presumably because it has been able to learn from such a large amount of data. Another possible reason is that humans did not spend the time or mental energy to perform as well as they could. To examine these possibilities, we performed a second evaluation with the same set of humans. In this evaluation, instead of having to consider all possible repairs – up to 100 candidates – the humans only had to decide between the four "most likely" repair candidates. These candidates were generated by taking the top four predictions from the SSC model (or the top three and the correct repair), shown in random order. In this evaluation, humans achieved 76% accuracy, which shows that the low performance of humans in the full task is due to the mental energy required, rather than lack of context or code understanding. We believe that these evaluations demonstrate that Real-Bug Test is a challenging set and that the accuracy achieved by the SSC model is empirically strong.

# 7 Analysis and Discussion

Our first goal is to conceptually understand at what "level" the model was able to generalize to new snippets. Although the hidden activations of the neural network model are not directly interpretable, we can attempt to interpret the latent model space using nearest neighbor retrieval on the hidden vectors $h_i$. The goal is to determine if the model is simply memorizing common $n$-grams, or if it is actually learning high-level repair concepts. Nearest neighbor retrieval for several test snippets are presented here:

|  | Example 1 | Example 2 | Example 3 |
|---|---|---|---|
| Test Snippet | ```
def _create_project(self, pname):
    project = models.Project()
    project.name = pname
    stage = models.Stage()
    stage.project = project
    hook = hook_models.Hook()
    hook.stage = stage
    hook.project = pname → project
    return hook
``` | ```
def get_median(self):
    l1 = len(self.min_h)
    l2 = len(self.max_h)
    m = l1 + l2 - 1 / 2
    if (m == l2 - 1):
        return (-self.max_h[0])
    elif l1 → m == l2):
        return self.min_h[0]
    raiseException()
``` | ```
def setupRedirect(self, type):
    if (type < → == 'Express'):
        self.setup_express())
    else:
        self.setup_full()
``` |
| Training Nearest Neighbor | ```
def handle_hash(self, handle):
    prefix = self.get_payload()
    hash = self.get_hash(prefix)
    db_hash = handle.DBHash()
    db_hash.prefix = handle → prefix
    db_hash.hash = hash
    response = self.run(db_hash)
    return repsonse
``` | ```
def cmp_datetimes(f, s):
    p_f = parse_date(f)
    p_s = parse_date(s)
    if (p_f > p_s):
        return -1
    elif f → p_f == p_s):
        return 0
    else:
        return 1
``` | ```
def test_restart(daemon_setup):
    exec_add({'command': 'sleep 5'})
    r = send_command({'key': 0})
    assert (r['status'] > → == 'err')
``` |

In Example 1, we see the model is able to learn a high-level pattern "$y.x = x$". In Example 2 we see the pattern "`if` ($x$ $c_1$ $y$...) `elif` ($x$ $c_2$ $y$...)". In Example 3 we see the pattern "Strings usually use the equality (or inequality) operator." In all cases, the surface form of the training nearest neighbor is very different from the test snippet. From this, it appears that the SSC model is able to learn a number of interesting, high-level patterns which it uses to generalize to new data.

We next examined failure cases of the SSC model which a human evaluator was able to repair correctly. Here, the primary weakness of the model was that humans were able to better infer program intent by using variable names, function names, and string literals. One major fault in the current implementation is a lack of sub-word representation. For example, consider a repair of the expression "`dtypes.append($x$)`" where $x$ could be `dtype` or `syncnode`. It is easy for a human to infer that `dtype` is the more sensible choice even without deeper understand of the code. In future work we plan to explore character-level encoding of value strings so that lexical similarity can be modeled latently by the network.

We finally examined cases where the SSC model succeeded but the human evaluator failed. Generally, we conclude that the model's primary advantage was the sheer amount of data it was able to learn from. For example, consider the expression "`if (db.version_info <= 3)`". This may not be immediately suspicious to a human, but if we analyze the reference training data we can measure that the pattern "`if ($x$.version_info <= $y$)`" is 10 times less frequent than the pattern "`if ($x$.version_info < $y$)`". Intuitively, this makes sense because if a feature is added in version $y$, it is not useful to check $<= y$. However, the neural model is able to easily learn such probabilistic distributions even without deeper understanding of *why* they are true.

# 8 Conclusion

We presented a novel neural network architecture that allows specialized network modules to explicitly model different transformation types based on a shared input representation. When applied to the domain of semantic code repair, our model achieves high accuracy relative to a seq2seq baseline and an expert human evaluation. In our analysis of the results, we find that our system is able to learn fairly sophisticated repair patterns from the training data. In future work we plan to expand our model to cover a larger set of bug types, and ideally these bug types would be learned automatically from a corpus of real-world bugs. We also plan to apply the SSC model to other tasks.

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

## A   POOLED POINTER MODULE IMPLEMENTATION

Figure 2 provides a diagram of the pooled pointer network module.

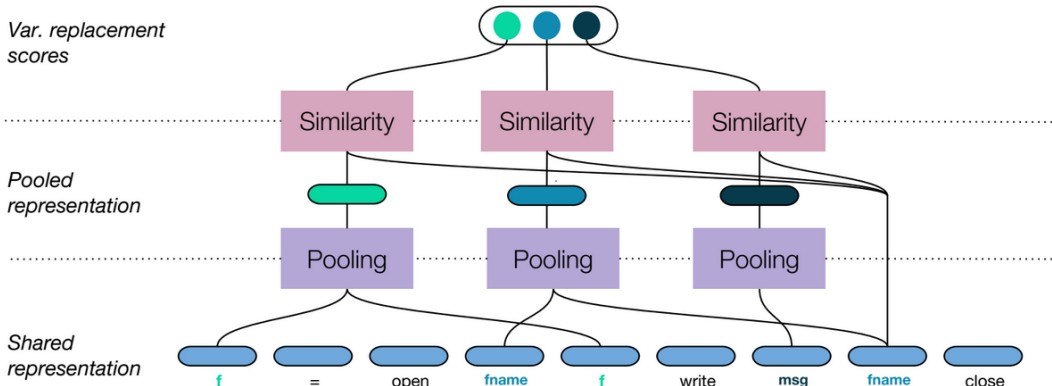

Figure 2: **Pooled Pointer Module**: The application of a pooled pointer module at a single time step, to predict the variable replacement scores for each potential replacement of the token *fname*. The input here is the per-token representation computed by the SHARE module. Representations for variable names are passed through a pooling module which outputs per-variable pooled representations. These representations are then passed through a similarity module, as in standard pointer networks, to yield a (dynamically-sized) output dictionary containing one score for each unique variable.

As described in Section 5.2, the pooling module consists of a projection layer followed by a pooling operation. For each variable $i$, its representation is computed by pooling the set of all its occurrences, $p_i$.

$$v_i = \mathrm{MaxPool}_{k \in p_i}(\tanh(Vh_k))$$

where $h_k$ denotes the representation computed by the SHARE module at location $k$.

The similarity module produces un-normalized scores for each potential variable replacement $i$. When applied at repair location $j$, it computes:

$$s_{ij} \;=\; \tanh(Wh_j) \cdot v_i$$

## B   EXAMPLES OF PREDICTIONS

We include the full set of system predictions for the Real-Bug Test set. We have made these available at `https://iclr2018anon.github.io/semantic_code_repair/index.html`.

## C   ADDITIONAL RESULTS

**Varying source code complexity** Figure 3 presents accuracy of the model across functions with varying numbers of repair candidates. While the repair accuracy decreases with the number of repair candidates, the model achieves reasonably high accuracy even for functions with over 100 repair candidates. Among functions with 101-150 repair candidates, the model accuracy is 73% for synthetically introduced bugs and 36% for real bugs.

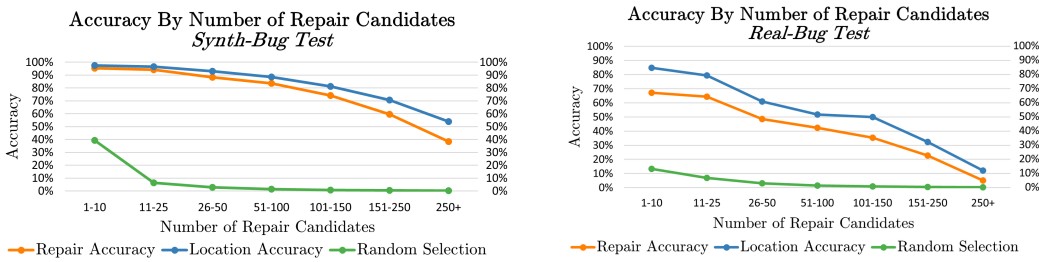

Figure 3: Results binned by number of repair candidates in the snippet

**Importance of AST structure** The Python abstract syntax tree is a rich source of semantic information about the tokens in a snippet. As described in Section 5.1, in addition to the original token string, we also include (1) the absolute position of the node in the AST, (2) the type of the node, and (3) the relationship between the node and its parent. To test the model's reliance on this information, we present ablation results over these additional feature layers below in Table 2.

We see that using information from the AST provides a significant performance gain. Still, even when only using the surface form values, the SSC model outperforms the attentional sequence-to-sequence baseline by a large margin (78.3% repair accuracy compared to 26% for the sequence-to-sequence model).

|  | Repair Accuracy | Location Accuracy |
|---|---|---|
| All | 87.1% | 91.3% |
| No Pos. | 86.8% | 91.1% |
| No Pos., Rel. | 85.7% | 90.9% |
| No Pos., Rel., Val. (Type Only) | 80.9% | 86.7% |
| Value Only | 78.3% | 84.3% |

Table 2: Results on *Synth-Bug Test* with ablation on different token types from the input AST representation.

