# OpenReview forum: "Semantic Code Repair using Neuro-Symbolic Transformation Networks"
_ICLR.cc/2018/Conference — Invite to Workshop Track_

### Official Review · AnonReviewer1 · 2017-11-27
**This paper presents a neural network architecture for program repair. Although this paper contains several strong points, the weaknesses of this paper are also very obvious.**

**Rating:** 4
**Confidence:** 4

**Review:**

This paper presents a neural network architecture consisting of the share, specialize and compete parts for repairing code in four cases, i.e., VarReplace, CompReplace, IsSwap, and ClassMember. Experiments on the source codes from Github are conducted and the performance is evaluated against one sequence-to-sequence baseline method.

Pros:

* The problem studied in this paper is of practical significance.
* The proposed approach is technically sound in general. The paper is well-written and easy to follow.

Cons:

* The scope of this paper is narrow. This paper can only repair the program in the four special cases. It leads to a natural question that how many other cases besides the four? It seems that even if the proposed method works pretty well in practice, it would not be very useful since it is effective to only 4 out of a huge number of cases that a program could be wrong.

* Although the proposed architecture is specially designed for this problem, the components are a straight-forward application of existing approaches. E.g., The SHARE component that using bidirectional LSTM to encode from AST has been studied before and the specialized network has been studied in (Andreas et al., 2016).  This reduces the novelty and technical contribution of this paper.

* Many technical details have not been well-explained. For example, how to determine the number of candidates m, since different snippets may have different number of candidates? How to train the model? What is the loss function?

* The experiments are weak. 1) the state-of-the-art program repair approaches such as the statistical program repair models (Arcuri and Yao, 2008) (Goues et al., 2012), Rule-Based Static Analyzers (Thenault, 2001) (PyCQA, 2012) should be compared. 2) the comparsion between SSC with and Seq-to-Seq is not fair, since the baseline is more general and not specially crafted for these 4 cases.

---

> ### Author Response · Authors · 2017-12-26
> **Response**
>
> Thanks for the review and questions. In our response, we briefly explain why the 4 classes of bugs we consider in this work are actually quite broad, and why other state-of-the-art program repair techniques are not applicable in our setting of identifying and repairing the programs without having access to test cases.
>
> Q. Scope of the paper is narrow and considers only 4 classes of bugs?
>
> First, we would like to point out that the 4 classes of semantic bugs that we chose were based on an extensive analysis of common classes of errors that programmers make, and which experienced programmers can potentially fix by only observing the program syntax without having access to any test cases or runtime information.
>
> Second, the 4 classes we consider (VarReplace, CompReplace, IsSwap, and ClassMember) are very broad classes of bugs. Our test set (https://iclr2018anon.github.io/semantic_code_repair/index.html)  shows both the prevalence and extreme diversity of these classes of bugs.
>
> Finally, there are other recent papers such as (http://bit.ly/2Dh7Qx8) that use models to identify only 1 class of bugs “Variable Misuse” that is similar to our VarReplace class.
>
> Q. how to determine the number of candidates m, since different snippets may have different number of candidates? How to train the model? What is the loss function?
>
> For each snippet, our model first uses the SHARE module to emit a d-dimensional vector for an AST node of the snippet, which are then encoded using a bi-LSTM to compute a shared representation H. Next, for each repair type, the SPECIALIZE module uses H and either an MLP or a Pooled Pointer module to produce an un-normalized scalar score for each of the m repair candidates. For a given snippet, we first identify the possible repair locations based on our 4 classes. For each repair location, the m candidates are computed depending on the AST node class. For example, if the repair location is of type comparison operator, it will consists of m=7 repair candidates, where 7 is the number of comparison operators we consider (==, <=, >=, <, >,!=,No-op). Similarly, for IsSwap and ClassMember there are 2 choices per location and a No-op. For VarReplace, the corresponding candidates for a variable node is computed by considering every other variable node defined in the program. Finally, a separate softmax is used for each candidate repair location to generate a distribution over all repair choices at that location (including No-Op).
>
> Since we train our model on a set of synthetically injected bugs, we know exactly for a given snippet which candidate repairs are applicable (if any). For each repair instance (snippet+repair location), we obtain a different training instance, and use the standard cross-entropy loss to get the softmax distribution as close as possible to the ground truth corresponding to the injected bug.
>
> Q. the state-of-the-art program repair approaches such as the statistical program repair models (Arcuri and Yao, 2008) (Goues et al., 2012), Rule-Based Static Analyzers (Thenault, 2001) (PyCQA, 2012) should be compared
>
> Please note that the state-of-the-art statistical approaches for program repair such as (Arcuri and Yao, 2008) and (Goues et al. 2012) use a set of test-cases to perform evolutionary algorithm to guide the search for program modifications. Our goal in this work is to automatically generate semantic repairs only looking at the program syntax without any test cases. This requirement is important because it forces development of models which can infer intended semantic purpose from source code before proposing repairs, as a human programmer might.
>
> The general rule based static analyzers only consider shallow syntactic errors and do not consider the class of semantic errors we are tackling in this work, so they would not produce any results.
>
> Q. the comparsion between SSC with and Seq-to-Seq is not fair, since the baseline is more general and not specially crafted for these 4 cases.
>
> Attention based seq-to-seq trained on the same training set is the closest state of the art model previously proposed in recent syntactic program repair approaches (Gupta et. al. AAAI 2017 and Bhatia et. al. 2016).
>
>
> Please let us know if there are any more clarifications that might be needed. We would like to reinforce this again that one of the goals of our work is to develop new neural models that are able to identify a rich class of semantic bugs without any test cases.

---

### Official Review · AnonReviewer3 · 2017-11-28
**Interesting and challenging application with impressive results, but maybe a bit narrowly focused in its scope.**

**Rating:** 6
**Confidence:** 4

**Review:**

This paper introduces a neural network architecture for fixing semantic bugs in code.  Focusing on four specific types of bugs, the proposed two-stage approach first generates a set of candidate repairs and then scores the repair candidates using a neural network trained on synthetically introduced bug/repair examples. Comparing to a prior sequence-to-sequence approach, the proposed approach achieved dominantly better accuracy on both synthetic and real bug datasets. On a real bug dataset constructed from GitHub commits, it was shown to outperform human.

I find the application of neural networks to the problem of code repair to be highly interesting. The proposed approach is highly specialized for the specific four types of bugs considered here and appears to be effective for fixing these specific bug types, especially in comparison to the sequence-to-sequence model based approach.  However, I was wondering whether limiting the output choices (based on the bug type)  is going a long way toward improving the performance compared to seq-2-seq, which does not utilize such output constraints.  What if we introduce the same type of constraints for the seq-2-seq model? For example, one can simply modifying the decoding process such that for locations that are not in the candidate set, the network simply  makes no change, and for candidate-repair locations, the output space is limited to the specific choices provided in the candidate set.  This will provide a more fair comparison between the different models.
Right now it is not clear how much of the observed performance gain is due to the use of these constraints on the output space.

Is there any control mechanism used to ensure that the real bug test set do not overlap with the training set? This is not clear to me.

I find the comparison result to human performance to be interesting and somewhat surprising. This seems quite impressive.  The presented example  where human makes a mistake but the algorithm is correct is informative and provides some potential explanation to this. But it also raises a question. The specific example snippet could be considered to be correct when placed in a different context.  Bugs are context sensitive artifacts. The setup of considering each function independently without any context seems like an inherent limitation in the types of bugs that this method could potentially address.  Some discussion on the limitation of the proposed method seems to be warranted.




Pro:
Interesting application
Impressive results on a difficult task
Nice discussion of results and informative examples
Clear presentation, easy to read.

Con:
The comparison to baseline seq-2-seq does not seem quite fair
The method appears to be highly specialized to the four bug types. It is not clear how generalizable it will be to more complex bugs, and to the real application scenarios where we are dealing with open world classification and there is not fixed set of possible bugs.

---

> ### Author Response · Authors · 2017-12-26
> **Review Response**
>
> Thanks for the helpful comments and suggestions.
>
> Q. What if we add additional constraints on the output choices for seq2seq decoder to only candidate locations?
>
> This constraint of only modifying the candidate locations is implicitly provided in our training set, where only bugs at candidate locations are provided and the remaining code is copied. When we analyze the baseline results, the seq2seq network is quite good at learning such a constraint of only modifying the candidate locations and it gets the right repair about 26% of cases (and 40% with some additional modifications). The remaining cases for which it makes mistakes in suggested repairs, it either predicts the wrong repair or chooses the wrong program location, but it performs such modifications only at the candidate locations, i.e. it already learns the constraint to only modify the candidate locations.
>
> Q. Is there any control mechanism used to ensure that the real bug test set do not overlap with the training set?
>
> For the synthetic bug dataset (real code with synthetically injected bugs), we partition the data into training, test, and validation at the repository level, to eliminate any overlap between training and test. Moreover, we also filter out any training snippet which overlapped with any test snippet by more than 5 lines.
> The real bug dataset (real code with real bugs) was obtained by crawling a different set of github repositories from the ones used in training. We also ensure there is no overlap of more than 5 lines with training programs.
>
> Q. Discussion about limitation of this work regarding not leveraging the context in which snippets are being used.
>
> Thanks for the suggestion. We will add a new paragraph regarding this limitation and future work. Yes, our current model is trained on a dataset where we extracted every function from each Python source file as a code snippet. Each snippet is analyzed on its own without any surrounding context. Adding more context regarding usage of functions in larger codebases would be an interesting future extension of this work, which will involve developing more scalable models for larger codebases.
>
> Q. Specialized to only 4 classes of errors?
> First, we would like to point out that the 4 classes of semantic bugs that we chose were based on an extensive analysis of common classes of errors that programmers make, and which experienced programmers can potentially fix by only observing the program syntax without having access to any test cases or runtime information.
>
> Second, the 4 classes we consider (VarReplace, CompReplace, IsSwap, and ClassMember) are very broad classes of bugs. Our test set (https://iclr2018anon.github.io/semantic_code_repair/index.html)  shows both the prevalence and extreme diversity of these classes of bugs.
>
> Finally, there are other recent papers such as (http://bit.ly/2Dh7Qx8) that introduce new models to identify only 1 class of bugs “Variable Misuse” that is similar to our VarReplace class.

---

### Official Review · AnonReviewer2 · 2017-12-02
**Cool application of neural nets to bug repair, but only in 4 special cases**

**Rating:** 6
**Confidence:** 4

**Review:**

This paper describes the application of a neural network architecture, called Share, Specialize, and Compete, to the problem of automatically generating big fixes when the bugs fall into 4 specific categories. The approach is validated using both real and injected bugs based on a software corpus of 19,000 github projects implemented in python. The model achieves performance that is noticeably better than human experts.

This paper is well-written and nicely organized. The technical approach is described in sufficient detail, and supported with illustrative examples. Most importantly, the problem tackled is ambitious and of significance to the software engineering community.

To me the major shortcoming of the model is that the analysis focuses only on 4 specific types of semantic bugs. In practice, this is a minute fraction of what can actually go wrong when writing code. And while the high performance achieved on these 4 bugs is noteworthy, the fact that the baseline compared against is more generic weakens the contribution. The authors should address this potential limitation.  I would also be curious to see performance comparisons to recent rule-based and statistical techniques.

Overall this is a nice paper with very promising results, but I believe addressing some of the above weaknesses (with experimental results, where possible) would make it an excellent paper.

---

> ### Author Response · Authors · 2017-12-26
> **Response**
>
> We thank the reviewer for the helpful comments and suggestions.
>
> Q. Only 4 classes of semantic bugs?
>
> First, we would like to point out that the 4 classes of semantic bugs that we chose were based on an extensive analysis of common classes of errors that programmers make, and which experienced programmers can potentially fix by only observing the program syntax without having access to any test cases or runtime information.
>
> Second, the 4 classes we consider (VarReplace, CompReplace, IsSwap, and ClassMember) are very broad classes of bugs. Our test set (https://iclr2018anon.github.io/semantic_code_repair/index.html)  shows both the prevalence and extreme diversity of these classes of bugs.
>
> Finally, there are other recent papers such as (http://bit.ly/2Dh7Qx8) that use models to identify only 1 class of bugs “Variable Misuse” that is similar to our VarReplace class.
>
>
> Q. Baseline is generic and weak?
>
> Please note that in our problem setting, we do not have access to the set of test cases. Most of the previous semantic program repair techniques rely on the availability of a set of test cases to find a repair. The only input to our model is the buggy program (its Abstract syntax tree), and the model needs to learn to predict whether there is a semantic bug (amongst the 4 classes) present in the snippet and if yes, pinpoint the node location and suggest a repair. We chose the attentional seq-to-seq model because it is one of the common models that has previously been used in recent literature for syntactic program repair (Gupta et. al. AAAI 2017 and Bhatia et. al. 2016).

---

### Author Response · Authors · 2017-12-26
**Generality of the 4 repair cases?**

We thank the reviewers for their helpful comments and feedback. It seems although the reviewers liked our neural network architecture for semantic program repair, there is a common concern regarding the generality and scope of the 4 classes of bugs we selected for evaluation. We are explaining this concern in a separate comment just to reinforce the fact that the 4 classes we consider are actually quite general and cover a large number of program bugs in our exploratory study of github codebases, especially compared to other recent work that only considers 1 class (out of our 4 classes) and show its prevalence in other codebases.

First, we selected the 4 classes of semantic bugs based on an extensive analysis of popular Python codebases on github to identify common classes of errors that programmers make, and using the following criterion “Bugs
that can be identified and fixed by an experienced human programmer, without running the code or
having deep contextual knowledge of the program.” This requirement of not having test cases is important because it forces development of models which can infer intended semantic purpose from source code before proposing repairs, as a human programmer might, and is also a great real-world test bed for developing models of understanding source code. Note that this requirement also disallows using majority of recent statistical semantic program repair techniques that relies on the availability of test cases.

Second, the 4 classes we consider (VarReplace, CompReplace, IsSwap, and ClassMember) are very broad classes of bugs. Our test set (https://iclr2018anon.github.io/semantic_code_repair/index.html)  shows both the prevalence and extreme diversity of these classes of bugs.

Finally, there are other recent papers such as (http://bit.ly/2Dh7Qx8) that use models to identify only 1 class of bugs “Variable Misuse” that is similar to our VarReplace class.

---

### Decision · Program_Chairs · 2018-01-29
**ICLR 2018 Conference Acceptance Decision**

**Decision:**

Invite to Workshop Track

**Comment:**

To summarize the pros and cons:

Pro:
* Interesting application
* Impressive results on a difficult task
* Nice discussion of results and informative examples
* Clear presentation, easy to read.

Con:
* The method appears to be highly specialized to the four bug types. It is not clear how generalizable it will be to more complex bugs, and to the real application scenarios where we are dealing with open world classification and there is not fixed set of possible bugs.

There were additional reviewer complaints that comparison to the simple seq-to-seq baseline may not be fair, but I believe that these have been addressed appropriately by the author's response noting that all other reasonable baselines require test cases, which is an extra data requirement that is not available in many real-world applications of interest.

This paper is somewhat on the borderline, and given the competitive nature of a top conference like ICLR I feel that it does not quite make the cut. It is definitely a good candidate for presentation at the workshop however.